# Machine Learning for Optical Motion Capture-Driven Musculoskeletal Modelling from Inertial Motion Capture Data

**DOI:** 10.3390/bioengineering10050510

**Published:** 2023-04-24

**Authors:** Abhishek Dasgupta, Rahul Sharma, Challenger Mishra, Vikranth Harthikote Nagaraja

**Affiliations:** 1Doctoral Training Centre, University of Oxford, 1–4 Keble Road, Oxford OX1 3NP, UK; 2Laboratory for Computation and Visualization in Mathematics and Mechanics, Institute of Mathematics, Swiss Federal Institute of Technology Lausanne, 1015 Lausanne, Switzerland; 3Department of Computer Science & Technology, University of Cambridge, 15 J.J. Thomson Ave., Cambridge CB3 0FD, UK; 4Natural Interaction Laboratory, Department of Engineering Science, Institute of Biomedical Engineering, University of Oxford, Old Road Campus Research Building, Oxford OX3 7DQ, UK

**Keywords:** feed-forward neural network, gated recurrent unit, inertial motion capture, linear model, long short-term memory, machine learning, musculoskeletal modelling, optical motion capture, recurrent neural network, upper extremity

## Abstract

Marker-based Optical Motion Capture (OMC) systems and associated musculoskeletal (MSK) modelling predictions offer non-invasively obtainable insights into muscle and joint loading at an in vivo level, aiding clinical decision-making. However, an OMC system is lab-based, expensive, and requires a line of sight. Inertial Motion Capture (IMC) techniques are widely-used alternatives, which are portable, user-friendly, and relatively low-cost, although with lesser accuracy. Irrespective of the choice of motion capture technique, one typically uses an MSK model to obtain the kinematic and kinetic outputs, which is a computationally expensive tool increasingly well approximated by machine learning (ML) methods. Here, an ML approach is presented that maps experimentally recorded IMC input data to the human upper-extremity MSK model outputs computed from (‘gold standard’) OMC input data. Essentially, this proof-of-concept study aims to predict higher-quality MSK outputs from the much easier-to-obtain IMC data. We use OMC and IMC data simultaneously collected for the same subjects to train different ML architectures that predict OMC-driven MSK outputs from IMC measurements. In particular, we employed various neural network (NN) architectures, such as Feed-Forward Neural Networks (FFNNs) and Recurrent Neural Networks (RNNs) (vanilla, Long Short-Term Memory, and Gated Recurrent Unit) and a comprehensive search for the best-fit model in the hyperparameters space in both subject-exposed (SE) as well as subject-naive (SN) settings. We observed a comparable performance for both FFNN and RNN models, which have a high degree of agreement (r${}_{\mathrm{avg},\mathrm{SE},\mathrm{FFNN}}=0.90_{\pm 0.19}$, r${}_{\mathrm{avg},\mathrm{SE},\mathrm{RNN}}=0.89_{\pm 0.17}$, r${}_{\mathrm{avg},\mathrm{SN},\mathrm{FFNN}}=0.84_{\pm 0.23}$, and r${}_{\mathrm{avg},\;\mathrm{SN},\;\mathrm{RNN}}=0.78_{\pm 0.23}$) with the desired OMC-driven MSK estimates for held-out test data. The findings demonstrate that mapping IMC inputs to OMC-driven MSK outputs using ML models could be instrumental in transitioning MSK modelling from ‘lab to field’.

## 1. Introduction

Three-dimensional (3D) motion analysis helps numerically describe joint and segmental kinematics. It also serves to establish non-disabled and identify pathological movement patterns [1,2]. In recent decades, in silico musculoskeletal (MSK) models have offered the ability to non-invasively estimate in vivo information to facilitate objective, clinical decision-making to some extent [3,4]. In particular, the prediction of intersegmental/joint and muscle loading has proven immensely valuable in improving our understanding of the MSK system [3]. However, traditional MSK models can prove to be computationally expensive and typically require comprehensive input motion capture (mocap) data [3,5]. Besides, creating, scaling, and setting up MSK models for individual subjects and trials can be onerous and quite time-intensive. Furthermore, licenses for commercial MSK modelling software can be prohibitively expensive, and some of the open-source packages might be restrictive regarding features and/or applications. The routine use of this technique is limited in clinical settings due to implementation challenges and computational complexity [3,5,6].

A plethora of commercial and research mocap systems have been developed to facilitate the capture of 3D human movement data accurately and repeatably [7] that can subsequently help describe the MSK function. Nonetheless, each mocap system is accompanied by its pros and cons [2,7]. Anglin and Wyss [2] noted that stereophotogrammetric (or optoelectronic) techniques with passive (retro-reflective) markers seem to be especially suited for upper-extremity motion analysis [8] since they have high accuracy, are non-invasive, and typically do not affect task execution. Nevertheless, the primary drawbacks of (marker-based) Optical Motion Capture (OMC) systems include marker occlusion (i.e., line-of-sight requirement), high cost, being lab-based, etc. Conversely, Inertial Motion Capture (IMC) equipment is cost-effective, user-friendly, portable, and delivers full-body measurement abilities in the subject’s ecological setting or ‘field condition’. However, IMC systems might not be as accurate as OMC systems, which are widely regarded as the ‘gold standard’ for non-invasive mocap. Regardless, this ‘field ready’ option of IMC systems can be immensely valuable in low-resource settings and outdoor or real-world applications (such as sports and ergonomics). Wearable inertial sensors have evolved rapidly and are routinely used in different areas of clinical human movement analysis, e.g., gait analysis [9], instrumented clinical tests, stabilometry, daily-life activity monitoring, upper-extremity mobility assessment, and tremor assessment [10]. Such sensors have rapidly transitioned from use in constrained, lab-based practice to unsupervised and applied settings [11].

There is an increasing demand to democratise and shift objective MSK model-led investigations from ‘lab to field’ [12,13,14,15]. OMC data have been traditionally used to drive inverse-dynamics-based MSK models [16,17]; however, recent works have used IMC input for MSK models [18,19,20,21,22,23]. Notwithstanding, upper-extremity biomechanics research has lagged more than that involving the lower extremity [1,2,24] due to several barriers such as a lack of consensus on a standardised protocol, functional tasks, model complexity, tracking method, marker protocol, and performance metrics. Thus, it is imperative to undertake better modelling of upper-extremity biomechanics as they are integral to many activities of daily living.

Machine learning (ML) has received considerable attention in human motion analysis, especially in the last decade [13,14,15,25,26,27]. In particular, using supervised ML models to circumvent biomechanical models that are computationally expensive has become commonplace. Numerous studies have reported ML applications for generating in vivo insights (e.g., joint and muscle loading) from either OMC or IMC inputs [28,29,30,31,32,33,34,35,36,37,38] or markerless methods [27]. Among ML approaches, deep learning models, including Convolutional Neural Networks (CNN) and, increasingly, Recurrent Neural Networks (RNN), have become popular in lower-extremity biomechanical analyses [26]. Such ML techniques have multiple benefits over their MSK model-based counterparts—(i) although the ML model’s initial training is considerably time-intensive, making predictions corresponding to new data is computationally efficient, facilitating real-time applications; and (ii) they do not require the large amounts of data required for MSK models and help provide population-level insights [32].

Our previous work [30] estimated upper-extremity MSK model information solely from IMC input data. This study uses ML methods to map IMC input data to OMC-driven MSK outputs. It serves two purposes: (i) bypasses the dependency on the computationally expensive MSK model during the prediction phase; and, (ii) more importantly, obviates the requirement for an OMC system in the prediction phase but still estimates ‘OMC-quality’ MSK outputs, which are more accurate than IMC-derived MSK outputs because of IMC systems’ higher susceptibility to soft tissue artefacts [7,18,19,20]. It allows the prediction of (non-invasive) gold-standard OMC-quality MSK outputs for mobile/ambulatory applications (e.g., sports, ergonomics, recreational activities) or in low-resource settings using highly portable IMC systems. A similar study for the lower extremity has used simulated IMU inputs [39]. This work differs in using actual IMC inputs, which were synchronously collected with the OMC inputs, giving greater reliability in predicting OMC quality outputs and in its application to the upper extremity. As in that study, we have realised our objective using an ML approach. In particular, we used Feed-Forward Neural Networks (FFNN) and RNN to model the human upper-extremity motion analysis. RNNs are, in general, better than FFNNs for modelling time-series data, and in this work, we have explored the popular RNN cells (vanilla, Long Short-Term Memory [LSTM], and Gated Recurrent Unit [GRU]) in RNNs and bidirectional RNNs, in which the input time-series is processed in both forward and backward directions. It is important to note that such a model creation was only possible because of the unique dataset acquired for [20], where the upper-extremity motion for five non-disabled subjects was synchronously captured using both IMC and OMC systems. To the best of our knowledge, this is the first study (across upper and lower extremities) that uses the comparatively lower-quality (but highly portable) IMC data to estimate kinematic and kinetic variables on par with the gold-standard OMC-driven MSK outputs. Such an approach will be instrumental in bringing the technology from ‘lab to field’ where OMC-driven approaches are infeasible.

Finally, along with an entirely novel approach to map IMC inputs to OMC-driven MSK outputs, this study also addresses several other research gaps in this field. For instance, most research in this field lacks rigorous hyperparameter tuning when training ML architectures (highlighted in [25,30]) which is crucial for generalisable models. Moreover, we trained the selected ML models for two different settings—(i) subject-naive (all the trials from a subject are strictly either in test or training data), and (ii) subject-exposed (at least one trial from each subject is used in the training of the model). The subject-naive setting is the true check for the generalisability of ML models, which is often ignored. Moreover, from the motion analysis point of view, this study focuses on modelling the human upper extremity, a relatively under-studied but essential domain.

## 2. Materials and Methods

### 2.1. Data

Here, we have utilised the anonymised dataset from a previous study [20], approved by relevant Research Ethics Committees (Reference numbers: 16/SC/0051 and 14/LO/1975). This research involved five non-disabled adult male volunteers (Age = 22.80 ± 0.84 years; Weight = 66.25 ± 9.72 kg; Height = 1.75 ± 0.07 m; Body Mass Index (BMI) = 21.79 ± 3.49 kg/m${}^{2}$).

Each subject was instructed to perform three trials of the ‘Reach-to-Grasp’ task (in the *Forward* direction) at a self-chosen pace. A custom-built apparatus (Appendix A) was used to accommodate the anthropometric requirements of different subjects and enable task execution while seated [40,41].

Synchronised experimental data capture corresponding to task execution involved a 16-camera Vicon^TM^ T40S Series System (Vicon Motion Systems, Oxford, UK) for OMC data (sampled at 100 Hz) and a Full-body Xsens MVN Awinda Station (Xsens Technologies B. V., Enschede, The Netherlands) with 17 wireless IMU sensors for the IMC data (sampled at 60 Hz). Hence, we down-sampled the OMC-driven MSK outputs using cubic interpolation to match IMC input data’s sampling frequency (∼60 Hz). Passive retro-reflective markers (ø 9.5 mm) were placed on the participant’s skin using a double-sided hypoallergenic tape using the Plug-in Gait marker set [42]. Each IMU transmits data wirelessly to a master receiver in real-time and contains a gyroscope, magnetometer, 3D accelerometer, barometer, and thermometer [43]. Specifically, the IMU data from the gyroscope, magnetometer, and 3D accelerometer were used as inputs in the ML models. IMU sensors’ placement on the participants’ body segments and their setup, as well as calibration, was performed as per the Xsens MVN User Manual’s recommended protocol [44]. Data capture was synchronised between the two systems wherein the Xsens IMC system triggered the Vicon OMC system following the guidelines of Xsens with a specific trigger at the start and stop recording time [45]. Marker and sensor placement, as well as data capture for all subjects, were performed by the same tester (VHN) to remove inter-tester variability error.

OMC data was processed using Vicon Nexus^TM^ v.2.5 software [46] to clean and label the marker trajectories for MSK model inputs. The processed marker trajectories were exported in Coordinate 3D (.C3D) file format [47]. Meanwhile, Xsens MVN Analyze v.2018.0.0 software (Xsens Technologies B. V., Enschede, The Netherlands) [48] was employed for capturing the IMC data and exporting it as BioVision Hierarchy (.BVH) files for MSK model inputs where a stick-figure model was originally reconstructed. MSK modelling of the upper extremity was undertaken using AnyBody^TM^ Modeling System (AnyBody^TM^ Technology A/S, Aalborg, Denmark) separately for each of the mocap inputs as well as subject-specific anthropometric dimensions for scaling following these sources [18,41,49]. The ‘*Inertial MoCap model*’ and ‘*Simple Full-Body model*’ in the AnyBody Managed Model Repository (AMMR) v.2.1.1 (AnyBody^TM^ Technology A/S, Aalborg, Denmark) were adapted separately to compute the respective kinetic variables (i.e., joint moments, joint reaction forces, and muscle forces) and kinematic variables (i.e., joint angles) of interest.

Demographic and anthropometric characteristics (e.g., age, height, weight, gender) affect the amplitude of kinetic and kinematic outcome variables in clinical movement analysis, and if not corrected, individual disparities can act as confounding factors [50,51]. Therefore, the kinetic variables were normalised and reported per best practices in the field [50,51], which were also adopted and detailed in our earlier related study [30]. Besides, the joint angles sign convention is based on a related work [41].

Prior to data collection, each subject was provided with comprehensive instructions in addition to a few practice trials. Furthermore, the IMC mocap system underwent an initial calibration sequence that required volunteers to assume a neutral posture known as the ‘N-pose’ and take some steps forward and backwards to return to their original position. Next, participants performed a ‘Static’ trial by standing in a stationary neutral or anatomical position in the centre of the capture volume for the OMC system. After these steps, the Reach-to-Grasp task (three trials) with the subject’s dominant hand was performed. The experimental data captured from the IMC system and the OMC-data-driven MSK model estimations (i.e., kinematic and kinetic parameters) were employed for training the proposed ML models.

### 2.2. Supervised Learning

We developed ML models to obtain OMC-driven MSK outputs from IMC inputs for the human upper extremity as depicted in Figure 1. The IMC experimental data (refer Section 2.1 for details on input features) comprise 15 trials (obtained from five subjects with three trials of Reach-to-Grasp task each) with 483 input features [44]. Whereas OMC-driven MSK model output categories comprise time-series data of ten joint angles, four muscle forces, four muscle activations, twelve joint reaction forces, and ten joint moments. A different ML model is trained for each output category while using IMC experimental data as input. Prior to model training, the following feature transformations are applied, which are similar to those in our previous article [30]:
The time feature, normalised between zero and one, indicates the proportion of total time taken to complete the task at the subject’s chosen pace.Muscles are typically discretised into numerous muscle bundles in MSK models. MSK model outputs for muscle activations and muscle forces comprise features for the four considered superficial muscle groups, i.e., (a) Biceps Brachii, (b) Pectoralis major (Clavicle part), (c) Brachioradialis, and (d) Deltoid (Medial) [49]. The ‘maximum envelope’ of the tendon forces of specified bundles forming a certain muscle was calculated for further analysis. Muscle activation measures the force in a selected muscle relative to its strength.We have used many-to-one RNN architecture, which uses multiple previous inputs for an output; therefore, we transformed the time-series data into a sub-time-series of *t* frames by sliding across the original time-series in a step of one. Thus, the input of the RNN is t×483, where *t* was taken as 10.

To show the added utility of applying complex ML techniques, we first attempt a linear model to see if it provides satisfactory metrics. We use Pearson’s correlation coefficient (r) and Normalised root-mean-squared error (NRMSE) metrics to compare the effectiveness of our models. For subject-exposed (SE) models, linear models achieved ‘moderate’ to ‘strong’ performance in terms of r. For subject-naive (SN) models, linear models perform moderately in terms of r; however, the NRMSE is large for SN models (as listed in Appendix A and discussed in Section 3 and Section 4). Thus, more complex ML models are necessary.

Two neural network architectures were employed with normalised MSE as the cost function: (i) FFNN, depicted in Figure 2 shows a typical schematic diagram and (ii) RNN, depicted in Figure 3, shows a schematic architecture (containing at least one RNN cell) along with choices of RNN cells, including vanilla, LSTM, and GRU in both unidirectional and bidirectional (which processes the input time-series in both forward and backward directions) sense [52,53,54]. Here, it is worth noting that we have treated these choices in RNN cells as hyperparameters.

The validation accuracy was then used to tune FFNN and RNN hyperparameters through an exhaustive grid search in the space with 43,740 and 23,328 choices that include hyperparameters such as the number of nodes, weight initialization, batch-size, optimizer, hidden layers, activation function, learning rate, epoch, RNN cells, and dropout probability, as listed in Table 1 and Table 2 along with the optimal hyperparameters. The current hyperparameter exploration for FFNNs is informed by our previous related work [30].

### 2.3. Validation and Train–Test Split

Two approaches are commonly used to benchmark ML architectures that approximate MSK models: subject-exposed (SE) and subject-naive (SN) [55]. In the SE setting, all subjects’ trials are combined and then divided into a training and test set, guaranteeing every subject within the test set is represented by at least one trial within the training data. An ML model trained through this method displays superior performance compared to the SN setting, where subjects are exclusively in either test or training data. However, the downside of the SE model is that it is subject-specific and has poor generalisation capabilities [36].

In the train–test split for the SE case, two trials were randomly chosen as the held-out test set and the rest of the trials as the training set. We employed cross-validation to optimise the hyperparameters by randomly dividing the remaining 13 trials into two sets—the training set comprising 11 trials and the validation set containing two trials. In the SN scenario, one subject was chosen as the test subject randomly, and the others were treated as training data. Following this, cross-validation was conducted by randomly taking one subject in the validation set and the rest of the subjects in the training set.

### 2.4. Error Metrics

NRMSE (i.e., RMSE scaled by the standard deviation of the target variable) was used as the primary error metric as in [30,36], where the ground truth is the corresponding OMC-driven MSK output. Since RMSE (and thus NRMSE) is sensitive to outliers, we furthermore used Pearson’s correlation coefficients (r) between the outputs of the ML model and the MSK model as a secondary error metric.

## 3. Results

Figure 4 summarises mean r and NRMSE values (along with corresponding standard deviation) for each kinematic and kinetic output category (where the mean and standard deviation are reported over all output features in a particular category) for linear, FFNN, and RNN models with numerical data reported in the Appendix A. The linear model served reasonably well for SE settings (NRMSE${}_{\mathrm{avg}}=0.65_{\pm 0.40}$ and r${}_{\mathrm{avg}}=0.80_{\pm 0.21}$); however, compared to our previous work [30] mapping the IMC inputs (considered in this study) to the corresponding IMC-driven MSK outputs, the performance of the linear model has considerably reduced. In contrast, the performance of linear models for SN settings is considerably poor, particularly in terms of NRMSE${}_{\mathrm{avg}}=1.78_{\pm 1.46}$, while the trends in the MSK outputs are reasonably captured with r${}_{\mathrm{avg}}=0.71_{\pm 0.37}$.

Performing a rigorous hyperparameter search is essential to achieving good model fit and generalisability. In Table 1 and Table 2, we have listed the various hyperparameter choices (43,740 and 23,328 combinations) considered for FFNN and RNN models, respectively. Four-fold cross-validation was performed for each hyperparameter choice, and the model with the best validation accuracy was selected. The best-performing model (as listed as optimal hyperparameters in Table 1 and Table 2) was then re-trained on the training and validation data, with the final metrics reported for a held-out test dataset. We obtained the best performance for the FFNN architectures containing multiple hidden layers (greater than four in most cases) and wide layers containing up to 1800 nodes with the network initialised using either Xavier or Random normal initialization and parameters trained using the ReLU activation function (on hidden layers), Adam optimizer, dropout regularization, and a learning rate of 0.001 or 0.005. In the case of RNN models, we have treated various choices in RNN cells as hyperparameters with the observation that LSTM and B-LSTM cells perform the best for most cases, with Adam or RMSprop as the optimal optimizer. Interestingly, the best-fit RNN models are simpler than FFNN, with shallower and narrower networks containing only 1 or 2 RNN layers and 128 or 256 nodes in each layer. The same observation can be made based on the number of NN parameters reported in Appendix A, where best-fit RNN models have fewer parameters than the best-fit FFNN models. Also, the best-fit SN models contain more parameters than the SE models for the same MSK model output category.

The two NN models consistently outperformed the linear model in terms of both r${}_{\mathrm{avg}}$ and NRMSE${}_{\mathrm{avg}}$, with notable improvements in SN settings. The FFNN and RNN models’ performance is comparable, with one model performing better for certain MSK outputs, as shown in Figure 4. The best-fit FFNN models have shown a high correspondence with the OMC-driven MSK outputs in both SE (r${}_{\mathrm{avg}}=0.90_{\pm 0.19}$ and NRMSE${}_{\mathrm{avg}}=0.33_{\pm 0.18}$) and SN settings (r${}_{\mathrm{avg}}=0.84_{\pm 0.23}$ and NRMSE${}_{\mathrm{avg}}=0.85_{\pm 0.79}$) with comparable statistics for RNN models with r${}_{\mathrm{avg},\mathrm{SE}}=0.89_{\pm 0.17}$, NRMSE${}_{\mathrm{avg},\mathrm{SE}}=0.36_{\pm 0.19}$, r${}_{\mathrm{avg},\mathrm{SN}}=0.78_{\pm 0.23}$ and NRMSE${}_{\mathrm{avg},\mathrm{SN}}=0.87_{\pm 0.77}$). As can be expected, the SE models performed better than the SN models consistently across various ML model choices.

To further aid in the interpretation of results, Figure 5 and Figure 6 (along with Appendix A) show the correspondence between the NN-predicted outputs and the actual OMC-driven MSK outputs for the held-out test trial data. Most plots show excellent r (>0.85) with notable exceptions, particularly for SN settings, in the case of Elbow Mediolateral, Elbow Anteroposterior, Wrist Anteroposterior, and Trunk Mediolateral (joint reaction forces) and Trunk Flexion/Extension and Trunk Internal/External rotation (joint moments). For further information, we have tabulated all output features with maximum, minimum, and inter-quartile ranges for r${}_{\mathrm{avg}}$ and NRMSE${}_{\mathrm{avg}}$ in Appendix A. Moreover, in Appendix A, we have tabulated an RMSE${}_{\mathrm{avg}}$ (with units) that gives an absolute error in the MSK output predictions. Note that the interpretation of RMSE values comes with the caveat that different features cannot be compared directly due to their different scales.

## 4. Discussion

Here, ML methods are introduced to estimate OMC-driven MSK model outputs from experimental input data captured synchronously from a full-body IMC system. This helps bypass a computationally demanding MSK model and obtain the corresponding OMC-driven MSK outputs (non-invasive ‘gold standard’) while not needing the laborious and laboratory-bound OMC setup. The IMC system offers remarkable user-friendliness, high-quality data with nearly real-time delivery, and a relatively quick setup process. Compared to OMC systems, this method saves the time and effort to skillfully palpate bony landmarks to place markers and to handle the arduous task of markers’ post-processing or dealing with missing data.

We considered both subject-naive (SN) and subject-exposed (SE) scenarios—considering both are important [25], as the choice affects generalisability, and thus, model accuracy [36,55]. While SE models tend to perform better, they are not as generalisable to unseen subjects as SN models. SE models can incorporate subject-specific tendencies to better predict kinematic and kinetic variables corresponding to them. Various architectures were used, ranging from a simple linear model to deep NNs. While the linear model was adequate for SE, as expected, it could not reproduce results well in the SN case. Subsequently, we considered more complex architectures, such as FFNNs and RNNs. As opposed to our previous work [30], the discrepancies in metrics between linear models and both FFNN and RNN models are substantial, even for SE setting, which might be attributed to the presumably greater complexity in mapping OMC-driven MSK model output to IMC input in contrast to mapping IMC-driven MSK model output to the same IMC input. This is also corroborated by the observation that the FFNN models implemented in this study are more complicated (Table 1), generally requiring more hidden layers and neurons than a similar previous study [30].

In general, we obtained excellent correspondence in both FFNN and RNN predictions with the ground truth, i.e., OMC-driven MSK output in both SE and SN scenarios (Figure 4 and Appendix A), with some notable exceptions, in particular, for SN settings. Some of these issues could be attributed to the lack of adequate training data (we have N=5), which we hope to address in future work by adding subjects and considering synthetic data augmentation using biomechanical models [37,56]. Note that the held-out trials were randomly selected for SE and SN settings from data with both inter- and intra-subject variability. This variability can be visualised in MSK outputs. For example, refer to Appendix A for joint reaction forces estimated by the MSK model. Despite this variability and small sample size (N=5), the FFNN and RNN models show a high degree of agreement and low NRMSE values. The predictions could be further improved with a much larger sample size. Furthermore, the performance of FFNN and RNN models are comparable in this study despite the number of parameters in FFNN models being significantly greater than the corresponding RNN models (refer to Appendix A). Such RNN architectures capturing the underlying MSK function using lesser model parameters can be trained easily. It would be worth exploring their performance for a much larger sample size.

Previously, FFNN was shown to predict IMC-driven MSK model outputs with high accuracy using IMC data [30]. The performance of linear models and FFNNs is relatively modest in the present study compared to that in Sharma et al. [30]. Notably, both OMC and IMC mocap systems are prone to soft tissue artefacts [7,57] and instrumental errors [58] differently, which often leads to differences in pose estimation accuracy between the measurement systems (e.g., tracking long-axis rotations like shoulder internal/external rotation or elbow pronation/supination). In particular, IMC systems are more susceptible to soft tissue artefacts (i.e., the requirement of at least three passive markers placed on bony landmarks of a segment in an OMC system versus at least one IMU placed in the middle of the body segment in an IMC system). Soft tissue artefacts are typically task-dependent and joint-dependent and vary based on the joint’s degree of freedom [59]. Besides, the two motion tracking systems captured at different sampling frequencies might also have introduced complexity while mapping the considered IMC inputs and OMC-driven MSK outputs in the ML pipeline. However, the influence caused by these aspects needs to be scrutinised in future studies. Hence, the ML models’ performance in this study (versus our prior study [30]) needs to be interpreted within this context.

As a result of the experiment being conducted in a laboratory environment, good model performance does not necessarily imply ecological validity in terms of replication in a real-world context. Though the experiment task is a constrained, goal-oriented, cyclical Reach-to-Grasp task that is ubiquitous in everyday life [49], inferences from models trained on this task may not necessarily generalise to other settings. While a study [55] involving ML models on the lower extremity has shown generalisability across age and body morphologies (and thus gait); this may not be the case for the upper extremity because of a larger range of motion, higher degrees of freedom, and the three-dimensional nature of tasks [1,2]. The applicability of our model for *what-if* analyses involving the task with different object weights or differently shaped objects and other upper-extremity motions are yet to be explored.

So far, we have focused on this study’s utility in (i) bypassing the computationally expensive MSK model, *and* (ii) obtaining gold-standard OMC-driven MSK outputs from IMC inputs. However, more recent alternative methods such as inverse dynamics, inverse kinematics, and static optimisation can perform real-time data processing, thus alleviating one of the main issues with using MSK models, especially in field situations. While these are valuable alternatives to the MSK model, these approaches can only obtain real-time predictions by sacrificing accuracy and simplifying the model. For example, taking tendon compliance into account is essential in research that relies on the muscle–tendon interaction or stored elastic energy in tendons (e.g., in running) [60], and helps achieve human-like motion [61,62]. Such aspects are ignored in ‘real-time’ inverse dynamics. To achieve real-time performance, simplified models are numerically solved up to a small number of iterations, which decreases accuracy. Predictions improve if the models are allowed to run longer, such as in offline scenarios [63,64,65]. Thus these alternative methods cannot be generalised easily and encode assumptions about certain movements that make them limited. In contrast, ML models are flexible while offering accuracy and speed. The trade-off between speed and accuracy in modern non-ML-based alternatives to MSK models is an interesting question that we hope to return to in future work.

Our work relies on MSK models, and thus, it inherits some of the underlying limitations, such as not considering soft-tissue artefacts. While MSK models are a standard method to obtain kinematics and kinetics from motion capture, work is ongoing [3,60,66,67] in MSK model verification & validation (V&V). V&V aims to address the limitations of the MSK model in terms of accuracy concerning higher fidelity sources, such as bone-pin marker measurements [68], which minimise the impact of soft-tissue artefacts. Direct measurements [69,70], in general, and in vivo joint reaction forces [71,72,73] using endoprosthesis have been used to validate the MSK model. Any future improvements in the MSK model from such V&V efforts will consequently be reflected in ML models trained on MSK model outputs.

This work can be used as a starting point toward implementing a ‘lab-to-field’ approach by combining ML modelling with easily obtainable IMC data. This can be used for translational research that uses MSK model outputs on IMC data obtained in real-world settings, such as supermarkets and manual material handling [21,22,74,75,76], as a training corpus for an ensemble of ML models which can be used to facilitate clinical diagnosis rapidly. In addition to IMC data, recent work on markerless systems [77], such as two-dimensional recordings [35], to obtain mocap quality data provides another source of training data while being used without specialised equipment.

## 5. Study Limitations

Our study’s key limitation is the small sample size of N=5; nevertheless, the methodological framework introduced in this proof-of-concept study shows that it is possible to get ‘OMC-quality’ MSK model estimations reasonably from comparatively lower-quality IMC measurements and paves the way for more elaborate studies going forward. Many studies in this field have considered up to N=10 subjects [25].

In the scaled cadaver-based MSK model in the AnyBody Modeling System [16], the segment masses were deduced from the anthropometric data by Winter [78] and the subjects’ respective heights and weights. However, this approach might not be suitable for all body types, e.g., elderly or bariatric populations. It is acknowledged that the current sample population is fairly homogeneous; motion capture and MSK modelling for people with a higher BMI and/or body fat composition might be more susceptible to soft tissue artefacts. For instance, BMI has been found to affect upper-extremity kinematics in non-disabled subjects [79]. Hence, future studies should explore motion capture involving subjects with a wide range of BMI or body morphologies [80].

The magnitude of soft tissue artefacts for any movement is influenced by an individual’s anthropometric characteristics, such as, marker location, activity performed, and the instrumented segment [57,81]. Soft tissue artefacts are usually task-dependent and joint-dependent as well as vary based on the joint’s degree of freedom [59]. In particular, MSK model estimates are considerably affected by soft tissue artefacts [82] and body segment inertial parameters [83]. Furthermore, skin-to-skin contact may also influence results (e.g., contact between the upper arm and forearm during elbow flexion movement). Thus, measurements of obese subjects may require special consideration for these effects [84]. In summary, these are inherent limitations/legacy issues of current mocap systems and MSK models that affect their outputs, thereby affecting the ML predictions trained on them. Therefore, future studies should involve a larger and more heterogeneous sample size and training dataset to help with improved predictions and a more generalisable model.

Finally, even though our approach to mapping highly portable IMC inputs to (non-invasive gold-standard) OMC input-driven MSK outputs shows immense potential, the training of such ML models is limited to synchronised IMC and OMC data capture.

## 6. Recommended Future Work

We see relevant future work proceeding broadly along three lines of inquiry. We can incorporate our ML pipelines (presented here and earlier [30]) in relevant open-source tools [85,86] with a graphical user interface. These interfaces can assist users in estimating kinematic and kinetic variables of interest from mocap data without utilising an MSK modelling framework. In terms of utilising ML models on larger datasets and potentially working with open-access mocap data, an example of such data validated with an in vivo instrumented device is from the *Grand Knee Challenge* [69,70] for the lower extremity. Another prominent repository is the *CMU Motion Capture Dataset* [87]—one of the largest publicly available mocap datasets. We plan to expand the study to consider other ML architectures for upper-extremity biomechanical modelling, e.g., Gaussian Processes. This could be particularly beneficial in subject-naive settings where there is room for improvement. The benefit of Gaussian Processes compared to FFNNs and RNNs is that one automatically obtains information about uncertainty in predictions. Undertaking methodological variations could be considered, such as exploring the different NN frameworks or ML methods listed in [25,29,33] and evaluating their applicability to different tasks [88].

Although this study solely focuses on marker-based and inertial motion capture systems (amongst the most widely used mocap systems in research and clinical settings) [7], future studies should explore other vision-based technologies (e.g., Kinect sensors) in this context as they can be highly unobtrusive, cost-effective, user-friendly, and portable. Similarly, vision-based action recognition methods and pose estimation [27,77,89] should also be explored for transitioning biomechanical analyses beyond the laboratory in future studies. Similar to this study, we hypothesise regarding synchronously recording movements through surface markers (or markerless methods [27]) and capturing reference or ‘gold-standard’ kinematics through bone-pin markers, fluoroscopy, magnetic resonance imaging, etc., [68]. Unfortunately, among other barriers, these techniques can be highly invasive or cause radiation exposure when used routinely. Addressing this issue, sophisticated ML models could be trained in the future that map synchronised skin-mounted marker data to MSK model (kinematic and kinetic) outputs estimated using the comprehensive, high-fidelity (bone-pin marker or bio-imaging) experimental data [68]. This could be an essential technique leveraging ML benefits to tackle the issue of soft tissue artefacts [57,59] for similar tasks tracked through stereophotogrammetry using skin-mounted markers or markerless methods. Finally, future studies should also explore ML models’ robustness in the presence of artificial noise or contamination.

## 7. Conclusions

To the authors’ knowledge, this is the first-ever proof-of-concept or preliminary study (across lower and upper extremities) that uses experimental data captured from an IMC system to directly estimate OMC-driven MSK model outputs (kinematic and kinetic) using an ML approach, bypassing the need for a (lab-based) OMC system *and* a computationally expensive MSK model. This was achieved by following ML best practices, such as an extensive hyperparameter search and utilising dropout for regularization. We used FFNN and RNN architectures alongside linear models. Both FFNN and RNN architectures achieved comparable performance on both kinematic and kinetic variables across SE and SN settings with strong-to-excellent Pearson’s correlation coefficients and outperformed linear models. Our results demonstrate the effectiveness of these models, and more broadly, ML methods, in approximating MSK models, synergistically leveraging the benefits of both mocap systems.

Such ML models are promising alternatives to the MSK modelling process requiring OMC input data, which are superior to predictive models that use only IMUs. With their high accuracy, ease of use, and computational efficiency, these ML models (coupled with IMC inputs) have the potential to bring motion analysis and MSK modelling beyond the lab, particularly in resource-limited settings, as an affordable and accessible solution.

## Figures and Tables

**Figure 1 bioengineering-10-00510-f001:**
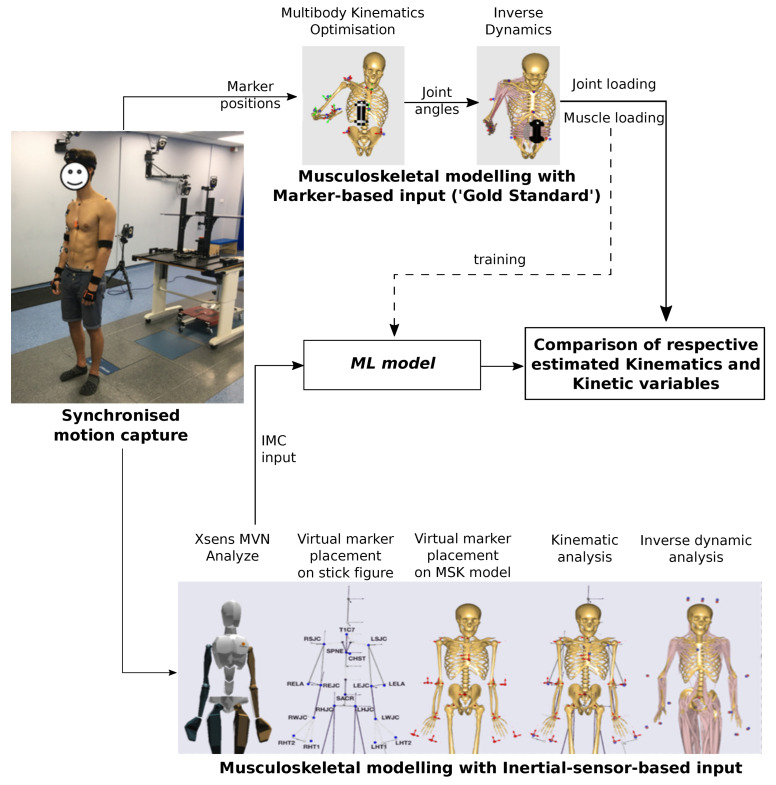
The machine learning (ML) pipeline used for musculoskeletal (MSK) modelling, which combines marker-based data (or Optical Motion Capture) and inertial-sensor-based data (or Inertial Motion Capture). The image has been adapted from [18,20,49].

**Figure 2 bioengineering-10-00510-f002:**
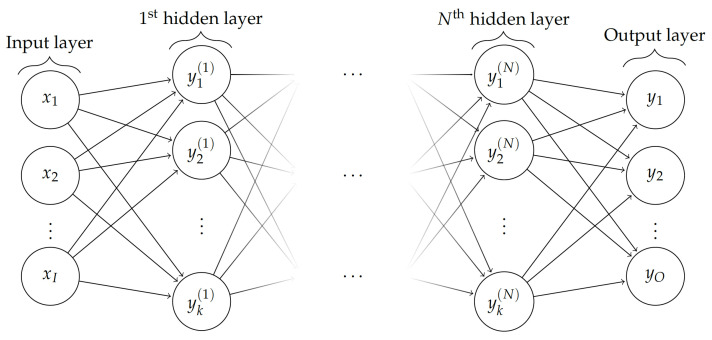
Typical Feed-forward Neural Network (FFNN) with *O* output nodes, *I* input nodes, and *N* hidden layers with *k* nodes/neurons each.

**Figure 3 bioengineering-10-00510-f003:**
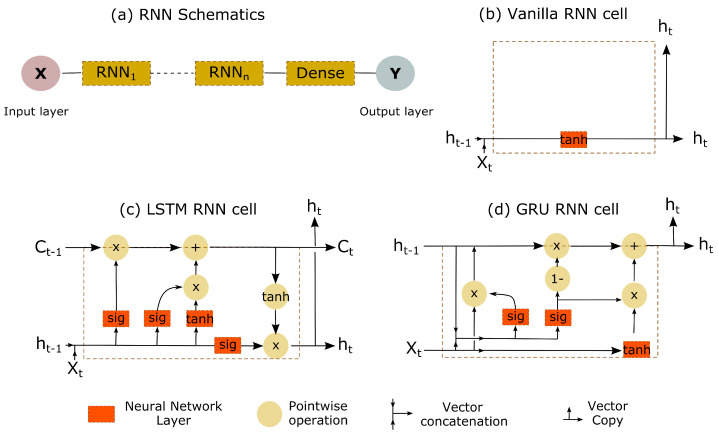
(**a**) Typical deep Recurrent Neural Network (RNN) architecture containing *n* RNN cells and a final dense layer [52,53,54]. The schematics of various RNN cells considered include: (**b**) vanilla RNN cell, (**c**) Long Short-Term Memory (LSTM) cell, and (**d**) Gated Recurrent Unit (GRU) cell. Note: $X_{t}$ is the input at any time *t*, $C_{t}$ is the cell state, and $h_{t}$ is the hidden state. Further note that the default activation function in RNN cells is tanh, while for recurrent steps in LSTM and GRU cells, the default recurrent activation function is sigmoid (sig).

**Figure 4 bioengineering-10-00510-f004:**
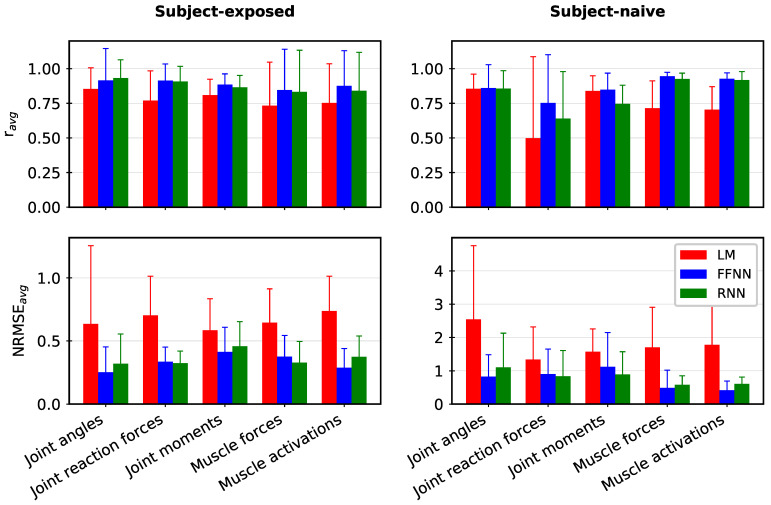
Average NRMSE values (NRMSE${}_{\mathrm{avg}}$) and average Pearson’s correlation coefficient (r${}_{\mathrm{avg}}$) for Linear Model (LM), Feed-Forward Neural Network (FFNN), and Recurrent Neural Network (RNN) predictions compared against the Musculoskeletal model outputs. The FFNN and RNN models consistently outperform the linear model. For a given output category, averaging is done over all output features and test trials (with the corresponding standard deviation shown as an error bar). See Appendix A for corresponding numerical data.

**Figure 5 bioengineering-10-00510-f005:**
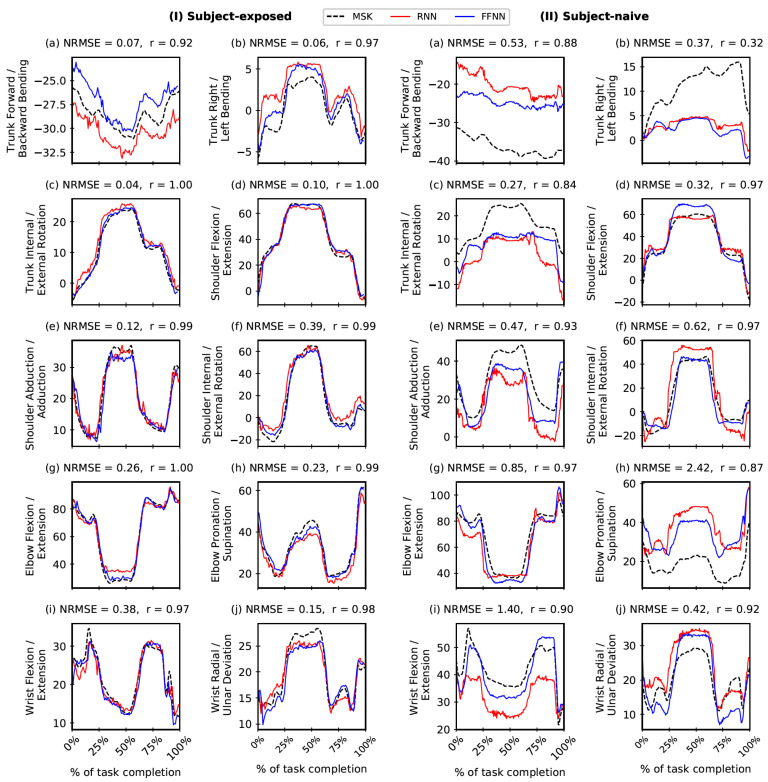
Comparing Feed-Forward Neural Network (FFNN) and Recurrent Neural Network (RNN) joint angles predictions (in degrees) with the corresponding outputs from the musculoskeletal (MSK) model. The comparison is shown for one trial in the held-out test data in Subject-exposed (**left**) and Subject-naive (**right**) settings. Note: Performance of FFNN and RNN are comparable (see Figure 4); in this figure, we report r and NRMSE values only for FFNN.

**Figure 6 bioengineering-10-00510-f006:**
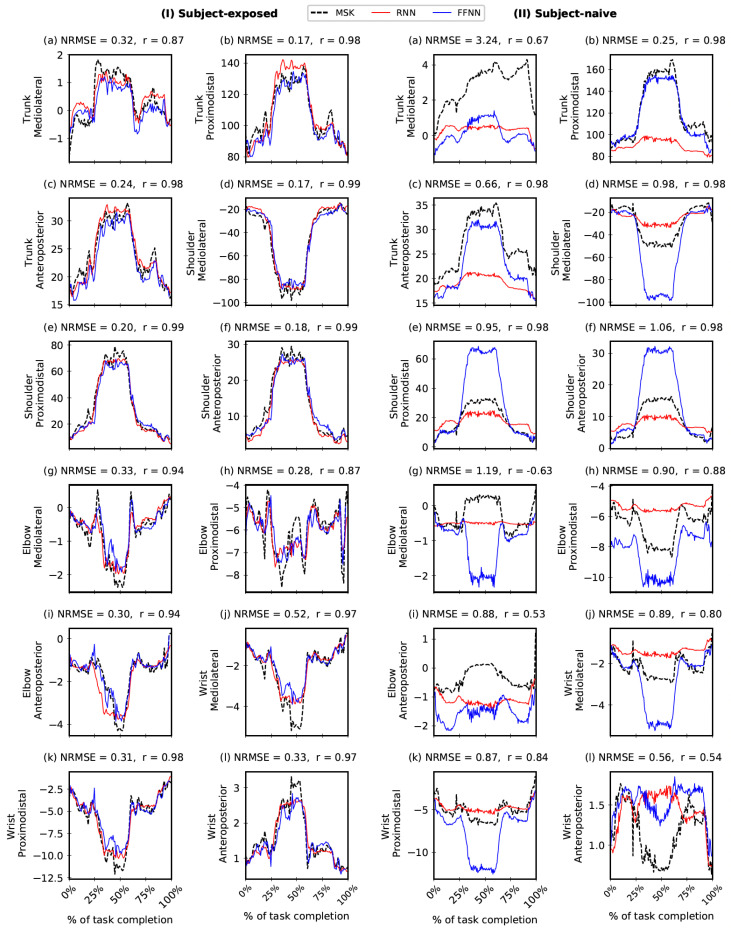
Comparing Feed-Forward Neural Network (FFNN) and Recurrent Neural Network (RNN) joint reaction forces predictions (in % Body Weight) with the corresponding outputs from the musculoskeletal (MSK) model. The comparison is shown for one trial in the held-out test data in Subject-exposed (**left**) and Subject-naive (**right**) settings. Note: Performance of FFNN and RNN are comparable (see Figure 4); in this figure, we report r and NRMSE values only for FFNN.

**Table 1 bioengineering-10-00510-t001:** Hyperparameters choices explored (43,740) for Feed-Forward Neural Network (FFNN) in subject-exposed and subject-naive settings and the optimal hyperparameters found for each MSK output category.

Output	Weight	Optimizer	Batch-Size	Epoch	Activation	Number of	Hidden	Learning	Dropout
	Initialization				Function	Nodes	Layers	Rate	Probability
**Hyperparameters explored**									
	He normal,	RMSProp,	64,	50,	ReLU,	200 to 1800	2, 4, 6,	0.001,	0, 0.2
	Random normal,	SGD,	256,	100,	sigmoid,	with increments	8, 10	0.005	
	Xavier normal	Adam	1028	200	tanh	of 200			
**Optimal hyperparameters**									
**Subject-exposed settings**									
Muscle forces	Random normal	SGD	64	50	ReLU	400	8	0.005	0.0
Muscle activations	Xavier normal	Adam	256	100	ReLU	1000	6	0.005	0.0
Joint angles	Random normal	Adam	256	200	ReLU	200	2	0.001	0.0
Joint reaction forces	Xavier normal	Adam	256	100	ReLU	200	8	0.005	0.0
Joint moments	Xavier normal	Adam	64	50	sigmoid	1400	2	0.001	0.2
**Subject-naive settings**									
Muscle forces	Xavier normal	SGD	64	200	ReLU	1200	8	0.005	0.2
Muscle activations	Random normal	Adam	256	200	ReLU	1200	6	0.001	0.2
Joint angles	Random normal	Adam	256	100	ReLU	800	4	0.005	0.0
Joint reaction forces	Random normal	Adam	256	50	ReLU	800	8	0.001	0.0
Joint moments	Xavier normal	Adam	64	50	sigmoid	1800	2	0.001	0.2

**Table 2 bioengineering-10-00510-t002:** Hyperparameters choices explored (23,328) for the Recurrent Neural Network (RNN) in subject-exposed and subject-naive settings and the optimal hyperparameters found for each MSK output category. In the RNN cell category, ‘B’ stands for Bidirectional cell (which processes the input time series in both forward and backward directions), LSTM is a Long Short-Term Memory cell, and GRU is a Gated Recurrent Unit cell.

Output	RNN Cell	Optimizer	Batch-Size	Epoch	Activation	Number of	RNN	Dropout	Learning
					Function	Nodes	Layers	Probability	Rate
**Hyperparameters explored for RNN**									
	Vanilla, LSTM,	Adam,	64,	50,	ReLU,	128,	1, 2,	0.1, 0.2	0.001,
	GRU, B-Vanilla,	SGD,	128,	100,	sigmoid,	256,	3, 4		0.005
	B-LSTM, B-GRU	RMSProp	256	200	tanh	512			
**Optimal hyperparameters**									
**Subject-exposed settings**									
Muscle forces	LSTM	RMSprop	64	100	tanh	256	1	0.1	0.001
Muscle activations	LSTM	RMSprop	64	50	sigmoid	128	3	0.1	0.001
Joint angles	B-LSTM	Adam	256	200	sigmoid	256	1	0.2	0.001
Joint reaction forces	LSTM	RMSprop	128	50	sigmoid	128	2	0.1	0.001
Joint moments	B-LSTM	RMSprop	64	100	sigmoid	256	1	0.1	0.001
**Subject-naive settings**									
Muscle forces	GRU	Adam	256	100	ReLU	512	3	0.2	0.001
Muscle activations	LSTM	RMSprop	128	100	tanh	512	2	0.1	0.001
Joint angles	B-LSTM	Adam	64	50	tanh	256	1	0.2	0.001
Joint reaction forces	LSTM	Adam	64	50	ReLU	128	4	0.1	0.001
Joint moments	GRU	Adam	128	100	sigmoid	256	2	0.2	0.005

## Data Availability

The codes used for implementing the ML models and analysing data can be found on https://github.com/rahul2512/MSK_ML_beta (accessed on 20 April 2023). Due to ethical and/or privacy concerns, the complete datasets used in this study are not publicly accessible; however, the test data is provided with the above scripts.

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
