# Peer review of "Machine Learning for Optical Motion Capture-Driven Musculoskeletal Modelling from Inertial Motion Capture Data"

_bioengineering, 2023, doi:10.3390/bioengineering10050510_

Round 1

Reviewer 1 Report

Dear Authors,

Nice done comparison of the two systems.

Question to132-134
You wrote the system has the following sensors in a IMU-Unit.

Line: Each IMU transmits data wirelessly to a master receiver in real-time and contains a gyroscope, magnetometer, 3D accelerometer, barometer, and thermometer

Please write, which ones were used for data collection during the measurement?

What is BLSTM?
See Table 2 and Supplement: Table 1 and 2

Author Response

Reviewer: 1

Comments and Suggestions for Authors:

Dear Authors,

Nice done comparison of the two systems.

Thanks so much for your kind feedback.

Question to132-134

You wrote the system has the following sensors in a IMU-Unit.

Line: Each IMU transmits data wirelessly to a master receiver in real-time and contains a gyroscope, magnetometer, 3D accelerometer, barometer, and thermometer

Please write, which ones were used for data collection during the measurement?

The IMU data from the gyroscope, magnetometer, and 3D accelerometer were used. This clarification has also been made in the Methods section.

What is BLSTM? See Table 2 and Supplement: Table 1 and 2

B-LSTM stands for ‘Bidirectional Long Short-Term Memory’. It consists of two RNN layers – one that processes the input sequence in the forward direction and another that processes it in the backward direction. This bidirectional processing allows the network to capture both past and future information about each element in the sequence, which can be beneficial for tasks that require understanding the context of each element, as we found in this case.

We have now also explained this in the main text and briefly in the table caption in the main article. Further, we have added this information explicitly in the Supplementary Information table captions.

Finally, we acknowledge that we had used the terms ‘BLSTM’ and ‘B-LSTM’ inconsistently/interchangeably; these have now been changed to B-LSTM throughout.

Reviewer 2 Report

The paper introduces an initial study on utilizing machine learning approaches to derive motion features from cheap inertial sensors. Overall the paper is well written and easy to understand. The basic approach is straightforward and illustrates an interesting and application relevant methodology. The general approach is technical sounds. Some drawbacks of the paper are:

1. The discussion of the solution focus strictly either between Marker based optical systems and inertial systems. Hence, other vision based technologies are ignored. This includes structured light using e.g. Kinect sensors, or other 2.5D vision systems and fully monocular (sparse and dense) pose reconstruction methods, which also have adopted to medical applications.

2. In the same direction the paper ignores recent methods from vision based action recognition which are based on very similar technical problems, i.e. direct regression from observed motion patterns into an output space.

2. The networks adopted are adequate, but more recent methods to sequence modelling (especially transformers) should be discussed.

3. It should be stated in the abstract, that the study is an initial proof of concept, especially because of the limited data basis.

4. Run time results and run time requirements should be given, since it is claimed that the ML solution is fast than the direct solution of the inverse kinematic model.

5. Figure 2 and 3 are simple textbook knowledge and provide no real information for this article.

6. The overall goal of generalization should be described more clearly. Should the final system generalize over subjects with standardized and limited coditions or should it in general be capable to analyse IMC data across motion patterns and subjects ?

Author Response

Reviewer: 2

Comments and Suggestions for Authors:

The paper introduces an initial study on utilizing machine learning approaches to derive motion features from cheap inertial sensors. Overall the paper is well written and easy to understand. The basic approach is straightforward and illustrates an interesting and application relevant methodology. The general approach is technical sounds. Some drawbacks of the paper are:

Many thanks for the kind feedback. We have now addressed your following comments and suggestions.

  1. The discussion of the solution focus strictly either between Marker based optical systems and inertial systems. Hence, other vision based technologies are ignored. This includes structured light using e.g. Kinect sensors, or other 2.5D vision systems and fully monocular (sparse and dense) pose reconstruction methods, which also have adopted to medical applications.

The discussion solely focuses on marker-based and inertial motion capture systems, two of the most widely used mocap systems in research and clinical settings. We agree that other vision-based technologies can and should be explored in this context in the future as they can be highly unobtrusive, cost-effective, user-friendly, and portable. 

Additional details have been added in the Future Work section in this regard.

  1. In the same direction the paper ignores recent methods from vision based action recognition which are based on very similar technical problems, i.e. direct regression from observed motion patterns into an output space.

Please see the response to the slightly-related comment above. We agree that vision-based action recognition methods (Mathis et al. 2020; Cronin 2021) should also be explored in future studies, although the current techniques are less suited for clinical applications, as their focus is on classification and are thus more robust to errors in the underlying pose estimation (Chen, C., Jafari, R. and Kehtarnavaz, N. (2017), “A survey of depth and inertial sensor fusion for human action recognition”, Multimedia Tools and Applications, Vol. 76 No. 3, pp. 4405-4425.; Shi, L.; Zhang, Y.; Cheng, J.; Lu, H. Skeleton-based action recognition with directed graph neural networks. In Proceedings of the 2019 IEEE/CVF Conference on Computer Vision and Pattern Recognition (CVPR), Long Beach, CA, USA, 15–20 June 2019; pp. 7904–7913.)

Additional details have been added in the Future Work section in this regard.

  1. The networks adopted are adequate, but more recent methods to sequence modelling (especially transformers) should be discussed.

Thank you for your feedback. However, it is important to note that this article focuses on exploring using neural networks as a proof of concept for mapping experimental Inertial Motion Capture (IMC) inputs to musculoskeletal model outputs derived by Optical Motion Capture (OMC) inputs. While we did not specifically discuss transformers, we agree that they are an important consideration for future work in this field. 

We wish to inform the reviewer that in a subsequent (and largely unrelated) study, we are conducting a comprehensive analysis and comparison of various neural network architectures, including transformers.

  1. It should be stated in the abstract, that the study is an initial proof of concept, especially because of the limited data basis.

The abstract has been modified as suggested.

  1. Run time results and run time requirements should be given, since it is claimed that the ML solution is faster than the direct solution of the inverse kinematic model.

In NN, the network's training is computationally expensive, but once trained, the time taken in a forward pass is negligible. In this case, the time required for the forward pass of one frame is approximately 0.13 ms to 0.5 ms for NN and 0.25 to 1.55 ms for RNN (variation in step time is for different NN models for various outputs and subject-exposed/subject-naive conditions). These time statistics are for using the networks on a laptop (2.4 GHz 8-Core Intel Core i9) and on a CPU (so no special requirement for GPUs).

The data is typically sampled at 100Hz, i.e., 1 frame per 0.01s. Or, in general, for real-time bio-feedback, the optimal latency (i.e., delay) is often considered to be 75ms (Kannape, O.A. and Blanke, O., 2013. Self in motion: sensorimotor and cognitive mechanisms in gait agency. Journal of Neurophysiology, 110(8), pp.1837-1847.) Thus, the NN networks are sufficiently faster to predict MSK outputs in real-time.

Traditional inverse kinematic models do not have real-time prediction ability and are relatively computationally expensive. However, as already described in the Discussion section, there are some simplified versions of MSK models (e.g., Van Den Bogert et al. 2013; Pizzolato et al. 2017a and 2017b) that provide real-time capabilities, albeit at a lower accuracy.

  1. Figure 2 and 3 are simple textbook knowledge and provide no real information for this article.

While Figures 2 and 3 are simple textbook knowledge, they serve the purpose of providing readers (especially part of the movement analysis community not familiar with AI/ML techniques) with visual representation and a better understanding of the neural network (NN) architectures being discussed in the article. Additionally, even for readers who are already familiar with these architectures, the figures can still be useful in clarifying the specific architectures being used in the present work. Therefore, while the figures may not contain any new information per se, they still have value in aiding the reader's comprehension of the article's content and making it more self-contained.

  1. The overall goal of generalization should be described more clearly. Should the final system generalize over subjects with standardized and limited conditions or should it in general be capable to analyse IMC data across motion patterns and subjects ?

Thank you for raising this point. We conclusively show that the models are generalisable for unseen subjects but with standardised and limited conditions. How generalisable the model will be over different motion patterns is difficult to comment on at this stage as we do not have data to test this. 

In some very limited previous works like Rane et al. 2019, NNs were shown to generalize well for new subjects of different ages and body morphology, often leading to different gait patterns. In another work, de Vries et al. trained NNs to predict shoulder joint reaction forces on several predefined upper-extremity movements while holding objects with different known masses. The authors found that the large variability in the tasks performed leads to poor NN approximations. Their results suggested that task-specific models should be designed over general models and question the reliability of such NN for new tasks. However, the NN trained was a simple shallow NN model with one hidden layer and 20 neurons.

Therefore it is not an easy question to address, especially defining what level of variability in tasks. In ongoing work, we are trying to address this question by testing the model for the same task but under different conditions. Our upcoming study should be able to answer this to some extent.

Reviewer 3 Report

The article entitled “Machine Learning for Optical Motion Capture-driven Musculoskeletal Modeling from Inertial Motion Capture Data” aims to propose a bypass allowing to make the direct link between measurement data derived from IMU and the evaluation of dynamic and kinematic components derived from musculoskeletal modeling by means of machine learning technique. On paper the idea is indeed very attractive, however the present study has shortcomings that require further information.

In the first place this paper is a continuation of another paper already published by the same authors. The goal is the same, the data is the same. Although the authors justify this study by the fact that it is given reference here are derived from measurement with a catching system with marker OMC and not by with IMUs as in the previous study. This justification needs to be further developed. Indeed, the IMU data also make it possible to do musculoskeletal modeling. This would mean that the approach here would actually correct the errors in the UMI measures to make them more compatible with a measure made in a OMC system.

The didactics of this paper need to be improved. The authors refer many methodological elements to their previous paper. This does not provide an overall view of the method. Indeed, I was very embarrassed not knowing the input format of the inputs and outputs of the neural networks. I recommend that this be clarified.

A musculoskeletal model requires not only kinematic data, but also anthropometric data and above all many parameters that make it possible to evaluate muscular mechanical actions. In this study, the population is very homogeneous, which marks this problem of parametrization of the musculoskeletal model. How do the authors envisage in the current architecture to take the intrinsic variations in the parametrization of musculoskeletal modeling?

As noted the number of subjects, but also the number of measurement sessions is insufficient. Given the number of parameters required for neural networks, I am not convinced that even 10 subjects would allow the entire space of possible parameters to be covered, given the very great possible variability between subjects, but also between musculoskeletal models. The risk of overfitting is not negligible.

Finally, I do not think that the correlation coefficients are a parameter that is robust enough to evaluate a difference in curve shape. In the results there are clearly some biases that are not quantified by this parameter.

In conclusion, this study presents original ideas, but its impact and validity have yet to be demonstrated. In a revised version, I recommend that the present study be considered as an exploratory approach to assess the feasibility of developing a bypass between kinematic data and the estimation of musculoskeletal components.

Author Response

Reviewer: 3

Comments and Suggestions for Authors:

The article entitled “Machine Learning for Optical Motion Capture-driven Musculoskeletal Modeling from Inertial Motion Capture Data” aims to propose a bypass allowing to make the direct link between measurement data derived from IMU and the evaluation of dynamic and kinematic components derived from musculoskeletal modeling by means of machine learning technique. On paper the idea is indeed very attractive, however the present study has shortcomings that require further information.

Many thanks for the kind feedback. We have now addressed your following comments and suggestions.

In the first place this paper is a continuation of another paper already published by the same authors. The goal is the same, the data is the same. Although the authors justify this study by the fact that it is given reference here are derived from measurement with a catching system with marker OMC and not by with IMUs as in the previous study. This justification needs to be further developed. Indeed, the IMU data also make it possible to do musculoskeletal modeling. This would mean that the approach here would actually correct the errors in the UMI measures to make them more compatible with a measure made in a OMC system.

The data used in this study only partly overlaps with the data used in our previous study (Sharma et al., 2022). The same is also true for the study motivation. The previous study only uses IMU sensor data (i.e., experimental data as input [features] and IMU-driven musculoskeletal model estimates as outputs [labels] in the machine learning model). However, in this study, we also have additional synchronised data from the OMC setup. To the best of our knowledge, this is the first study to estimate OMC-driven MSK quality kinematics and kinetic variables solely from experimental IMC input data.

MSK models driven by IMUs are not as accurate as those driven by more fine-grain marker data (Karatsidis et al., 2019; Konrath et al., 2019; Nagaraja et al., 2019). This is due to the inherent limitation of IMC systems regarding their pose estimation accuracy and influence of soft tissue artefacts relative to OMC systems. We have now highlighted this aspect in the Introduction to further develop the justification, as suggested.

The didactics of this paper need to be improved. The authors refer many methodological elements to their previous paper. This does not provide an overall view of the method. Indeed, I was very embarrassed not knowing the input format of the inputs and outputs of the neural networks. I recommend that this be clarified.

Thank you for pointing it out. We have further improved the Method section providing more details, particularly on inputs.

A musculoskeletal model requires not only kinematic data, but also anthropometric data and above all many parameters that make it possible to evaluate muscular mechanical actions. In this study, the population is very homogeneous, which marks this problem of parametrization of the musculoskeletal model. How do the authors envisage in the current architecture to take the intrinsic variations in the parametrization of musculoskeletal modeling?

In the scaled cadaver-based musculoskeletal model in the AnyBody Modeling System, the segment masses were deduced from the anthropometric data by Winter (2009) and the subjects’ respective heights and weights. However, this approach might not be suitable for all body types, e.g., elderly or bariatric populations.

We agree that our sample population is quite homogenous; motion capture and MSK modelling for people with a higher body mass index (BMI) and/or body fat composition—highly subject-specific—might be more susceptible to soft tissue artefacts (STA). Hence, future studies should explore motion capture involving subjects with a wide range of BMI and body morphologies (Horsak et al. 2018). This aspect has now been acknowledged in the Study limitations section. 

One possible implementation to internalise the variability due to anthropometric (scaling) data (e.g., subjects’ heights and weights) is to use it as part of the input features of the NNs instead of scaling the input and output features (as done in the current implementation), which we hope to address in future work. However, future studies should explore better ways of parametrisation of the MSK model for a highly heterogeneous population. Relevant details have now been added to the Study Limitations section.

As noted the number of subjects, but also the number of measurement sessions is insufficient. Given the number of parameters required for neural networks, I am not convinced that even 10 subjects would allow the entire space of possible parameters to be covered, given the very great possible variability between subjects, but also between musculoskeletal models. The risk of overfitting is not negligible.

Three trials are a standard approach for measurement sessions in movement analysis. Kadaba et al. (1989) recommended that one representative trial can generally be used for clinical decision-making. Previous studies of limited upper-limb motions have used one (Landry & Biden 2002) to five recordings (Mell et al. 2005). Based on the variation seen in preliminary data and considerations of muscle fatigue, it was decided that three trials would represent the typical movement of the task in this study.

Most studies using ML techniques in human movement biomechanics have dealt with up to ten subjects (Halilaj et al., 2018). Our dataset has sufficient training examples (~7000), as each training example is one sampling frame of the data. However, the current training dataset lacks diversity in terms of subjects to better capture the variability due to anthropometric measurements, which ultimately affects the generalisability of the neural network. For instance, if data from only 1 to 2 individuals are used to train a model, then the trained model will likely make poor predictions when tested on data from unseen subjects. In this work, we have thoroughly tested this by explicitly considering subject-naive settings, i.e., testing the model on unseen subjects, which shows reasonable results but certainly can be improved with more data. 

Furthermore, to mitigate the risks of overfitting, we have undertaken a rigorous hyperparameter search and used cross-validation and a held-out test set. Finally, as now highlighted in the Abstract and Conclusion (based on one of the other reviewer’s suggestions), this is a proof-of-concept study solely because of the small sample size.

Finally, I do not think that the correlation coefficients are a parameter that is robust enough to evaluate a difference in curve shape. In the results there are clearly some biases that are not quantified by this parameter.

We agree with your comment. Therefore, we have used additional metrics, including RMSE and NRMSE, which better quantifies such biases. The corresponding results are provided in Figure 4 and Supplementary Tables 1, 2, and 3. Furthermore, for one test trial (held-out test data), we have plotted the NN predictions with the ground truth (MSK model output) along with the various metrics in Figures 5, 6 and Supplementary Figures 4, 5, and 6, where one can visualise the curves and error in prediction in terms of various metrics.

In conclusion, this study presents original ideas, but its impact and validity have yet to be demonstrated. In a revised version, I recommend that the present study be considered as an exploratory approach to assess the feasibility of developing a bypass between kinematic data and the estimation of musculoskeletal components.

The suggested changes have now been made in the Conclusion section.